# Skeletal Deformities in Osterix-Cre;Tgfbr2^f/f^ Mice May Cause Postnatal Death

**DOI:** 10.3390/genes12070975

**Published:** 2021-06-25

**Authors:** Kara Corps, Monica Stanwick, Juliann Rectenwald, Andrew Kruggel, Sarah B. Peters

**Affiliations:** 1Department of Veterinary Bioscience, The Ohio State University College of Veterinary Medicine, Columbus, OH 43210, USA; corps.2@osu.edu (K.C.); rectenwald.3@osu.edu (J.R.); 2Division of Biosciences, The Ohio State University College of Dentistry, Columbus, OH 43210, USA; stanwick.10@osu.edu (M.S.); kruggel.5@osu.edu (A.K.)

**Keywords:** TGFbeta, osteochondral dysplasia, bone development, Cre recombinase, conditional knockout, off-target Cre, postnatal, mouse models, Osterix-Cre

## Abstract

Transforming growth factor β (TGFβ) signaling plays an important role in skeletal development. We previously demonstrated that the loss of TGFβ receptor II (Tgfbr2) in Osterix-Cre-expressing mesenchyme results in defects in bones and teeth due to reduced proliferation and differentiation in pre-osteoblasts and pre-odontoblasts. These Osterix-Cre;Tgfbr2^f/f^ mice typically die within approximately four weeks for unknown reasons. To investigate the cause of death, we performed extensive pathological analysis on Osterix-Cre- (Cre-), Osterix-Cre+;Tgfbr2^f/wt^ (HET), and Osterix-Cre+;Tgfbr2^f/f^ (CKO) mice. We also crossed Osterix-Cre mice with the ROSA26mTmG reporter line to identify potential off-target Cre expression. The findings recapitulated published skeletal and tooth abnormalities and revealed previously unreported osteochondral dysplasia throughout both the appendicular and axial skeletons in the CKO mice, including the calvaria. Alterations to the nasal area and teeth suggest a potentially reduced capacity to sense and process food, while off-target Cre expression in the gastrointestinal tract may indicate an inability to absorb nutrients. Additionally, altered nasal passages and unexplained changes in diaphragmatic muscle support the possibility of hypoxia. We conclude that these mice likely died due to a combination of breathing difficulties, malnutrition, and starvation resulting primarily from skeletal deformities that decreased their ability to sense, gather, and process food.

## 1. Introduction

Osterix (Osx or Sp7) is a zinc-finger family transcription factor required for osteoblast proliferation and differentiation [1]. It is expressed in early hypertrophic chondrocytes and inhibits chondrogenesis in favor of chondrocyte differentiation to regulate endochondral ossification [2,3,4]. Osterix-null embryos develop normal cartilage, but the osteoblasts fail to differentiate during endochondral ossification, resulting in a complete lack of bone tissue [5]. The Cre/loxP system is commonly used to knock out genes in specific cells and tissues at specific time points. Osx-GFP:Cre (Osx-Cre) mice have been used to target mineralized tissues, specifically bones and teeth, with the expression of the GFP/Cre fusion protein driven by the Osx promoter [6,7,8,9,10,11,12,13]. Osx-Cre activity was first detected in embryonic bones on embryonic day 12.5 (E12.5) and was maintained in bone and tooth progenitors through embryonic and postnatal development [7,14,15]. Breeding Osx-Cre mice with mice expressing a gene flanked by loxP sites generates a mouse with the target gene deleted in Osx-expressing cells.

The Osx-Cre mouse strain was initially used to characterize embryonic skeletal development [14] but has more recently been used to investigate postnatal development [6,7,8,9,10,11,12,13,16,17,18,19]. The Osterix-Cre transgene has been shown to cause defects in bone development that are generally overcome by 6 weeks of age [20,21,22]. Studies using reporter lines have demonstrated off-target Cre recombinase expression in stromal cells, adipocytes, and perivascular cells in the bone marrow, as well as in olfactory glomeruli cells and a subset of gastric and intestinal epithelial cells [12,23]. In spite of this off-target expression, many Osx-Cre/loxP mouse models fully mature, allowing investigation of skeletal development into adulthood [10,16,18].

Transforming growth factor β (TGFβ) plays a crucial role in regulating bone development during embryogenesis and postnatal life. This superfamily of proteins is activated when ligands bind to TGFβ receptor 2 (Tgfbr2), which then recruits TGFβ receptor 1 and subsequently forms a complex that initiates downstream signaling that transcriptionally activates tissue-specific differentiation factors. Tgfbr2 deletion in neural crest cells and early skeletal progenitors during embryogenesis results in embryonic lethality [24,25,26,27,28,29]. Late-stage Tgfbr2 deletion using the Osteocalcin-Cre/loxP model results in hypermineralization but not lethality [30,31]. We generated a conditional Tgfbr2 knockout mouse using the Osx-Cre/loxP model that survived approximately 4 weeks postnatally and demonstrated reduced growth and mineralization in bones and teeth [6,7]. The cause of death in these mice remained unknown, but reduced bone size alone seemed an unlikely cause of death even though several other Osx-Cre-driven gene deletions also caused postnatal lethality around 3–4 weeks of age [8,10,11,17]. We performed an extensive phenotypic pathological evaluation of tissues throughout the Osx-Cre+;Tgfbr2^f/f^ (CKO) mice and compared them to Osx-Cre+;Tgfbr2^f/wt^ (HET) and Cre negative (Cre-) controls. These reports revealed variations between individual mice and similarities between the HET and CKO mice that made them indistinguishable at times. We therefore report trends of phenotypic deformations seen in at least three CKO mice as compared to the Cre- mice, unless otherwise noted. We also utilized the ROSA26 mTmG reporter line to identify potential sites of off-target Cre expression non-skeletal organs. These detailed pathological findings are critical to understand the relevance of any Cre/loxP model in gene deletion investigations. In addition, this case report presents how accurate bone development in mice plays a crucial role in supporting their survival into and past postnatal stages.

## 2. Materials and Methods

### 2.1. Mice

All experiments with mice were approved by The Ohio State University Institutional Animal Care and Use Committee (IACUC), the number of which is 2020A00000023. CKO mice were described previously [6,7]. Mice were collected and analyzed on postnatal days 24–27 (P24–P27; see Figure 1). The Cre reporter strain Gt(ROSA)26SORtm4(ACTB-tdTomato,-EGFP) Luo/J (mTmG) was crossed with Osx-Cre mice to generate Osx-Cre+;ROSA26-mTmG (Osx-Cre;mTmG) mice with traceable Cre activity [32]. Reporter mice were collected and analyzed at 4–7 weeks. 

### 2.2. Pathology

The Comparative Pathology and Digital Imaging Shared Resource (CPDISR) of The Ohio State University Comprehensive Cancer Center performed all pathology procedures. An experienced necropsy technician (Ms. Rectenwald) performed standardized necropsies including an external examination, recording of body and major organ weights, and collection of all organs for histopathological assessment. Tibia length measurements and whole-body postmortem radiographs were additionally collected for a subset of the examined mice. Tissues were fixed for a minimum of 48 h in 10% neutral buffered formalin. Bony tissues, including the skull, were decalcified using Formical-2000 (StatLab, McKinney, TX, USA). Tissues were routinely processed for histopathology on a Leica Peloris 3 Tissue Processor (Leica Biosystems, Buffalo Grove, IL, USA), embedded in paraffin, sectioned at an approximate thickness of 4–5 micrometers, and batch stained with hematoxylin and eosin (H&E) on a Leica ST5020 autostainer (Leica Biosystems, Buffalo Grove, IL, USA) using a routine and quality-controlled protocol. Slides were evaluated by a board-certified comparative veterinary pathologist (Dr. Corps) using a Nikon Eclipse Ci-L Upright Microscope (Nikon Instruments, Inc., Melville, NY, USA). Representative photomicrographs were taken using an 18-megapixel Olympus SC180 microscope-mounted digital camera and cellSens imaging software (Olympus Life Science, Center Valley, PA, USA). 

### 2.3. Imaging of Off-Target Cre Expression in Osterix-Cre:ROSA26mTmG Reporter Mice

Soft tissue organs were collected from 4- and 7-week-old mTmG and Osx-Cre;mTmG mice. These were fixed in 4% paraformaldehyde for one hour, embedded, frozen in optimal cutting temperature compound, and sectioned at 15 µm thickness. Sections were stained with DAPI solution to mark nuclei and mounted for imaging using a Biotek Lionheart LX fluorescent microscope (*n* = 3 mTmG mice; 1 at 4 weeks and 2 at 7 weeks and *n* = 5 Osx-Cre;mTmG mice; 2 at 4 weeks and 3 at 7 weeks).

## 3. Results

We performed gross and microscopic pathological assessments on five Cre-, four HET, and nine CKO mice for a comprehensive view of the demonstrated phenotypes (Figure 1). Tissue analyses were performed on all mice in every genotype. Appendix A provides a comprehensive list of the lesions found in each genotype in all analyzed tissues. Our findings confirmed previous publications on this mouse model describing the axial and appendicular skeleton [4], maxillary and mandibular teeth [6], and some tissues known to exhibit off-target Cre expression (Figure 2) [12,23]. Conditional knockout mice were consistently smaller than comparable control littermates and age-matched controls (Figure 1), with short limbs and frequent facial bony abnormalities. Radiographic evaluation of a subset of these mice confirmed the previously reported abnormalities in the length of long bones, growth plates, and size and shape of the head [6,7], with shortening of the nose. Our evaluation also revealed additional bony alterations in the nasal passages, cribriform plate, and calvaria. HET mice were variable and sometimes indistinguishable from the CKO mice due to the range and severity of findings. No significant microscopic lesions were identified in most major organs, including the lungs, trachea, thymus, adipose, liver, spleen, kidneys, pancreas, salivary glands, esophagus, intestinal tract, haired skin, male and female reproductive tracts, sciatic nerve, thyroid glands, adrenal glands, spinal cord, and brain, excluding the olfactory bulb, where off-target Cre expression was confirmed (Figure 2G). Although there were no macro- or microscopic lesions or abnormalities identified, the stomach, intestines, liver, and lungs (Figure 2), as well as the kidneys, all had at least minor Cre expression. 

### 3.1. Visceral Muscle Injury Phenotypes Demonstrated in CKO Mice

Unexpectedly, we identified small multifocal areas of myocardial necrosis in either the left or right ventricles in six CKO and two HET mice. We did not observe any off-target Cre recombinase activity in the heart (Figure 2L), as previously reported [23]. Occasionally, cardiomyocytes in the area immediately surrounding the necrosis were mildly vacuolated, but more severe changes, including inflammation, were not observed (Figure 3A–D). Rare, slightly basophilic myofibers with multiple nuclei arranged in a linear row were observed in the skeletal muscle of the diaphragm and/or perivertebral muscles in six CKO mice and one HET mouse (Figure 3E,F). This morphology is most commonly associated with regeneration following injury, although no signs of such an inciting cause were observed.

### 3.2. Osteochondral Dysplasia in Appendicular Skeletons of CKO Mice

Histopathological changes were observed in numerous bones, including bones of the forelimbs and hind limbs, sternebrae, ribs, and the base of the skull, all of which were consistent with osteochondral dysplasia and failure or delay of endochondral ossification. These changes were most repeatable in the long bones of the forelimbs and hind limbs, including the humerus, radius, ulna, femur, tibia, fibula, and smaller bones of the distal joints and phalanges. In six CKO and two HET mice, changes in the metaphysis, physis, and epiphysis were also present, and several areas demonstrated reductions in bone matter, including thin, irregular cortices. Metaphyseal trabeculae were decreased in number, frequently irregular, decreased in bone matrix, and lined by fewer osteoblasts than expected. Physes were variable in thickness, with thickened areas frequently extending cartilaginous finger-like or frond-like projections into the epiphysis that were composed predominantly of disorganized prehypertrophic chondrocytes. Segments of the physes with more normal proportions of prehypertrophic and hypertrophic chondrocytes were thinner and more disorganized than similar structures in age-matched control mice. Nine CKO and three HET mice had misshapen epiphyses that demonstrated variable amounts of bone matrix, retained multifocal islands of cartilage, or were occasionally composed of an admixture of these with fibrous connective tissue. The articular cartilage was thickened in some CKO mice, while in others, it was thin and moderately sclerotic and hypereosinophilic. In all CKO and three HET mice, the articular cartilage was irregular on both surfaces of a joint (Figure 4A,B). Hematopoietic populations in affected bones were appropriate, with all expected lineages and levels of maturation in erythroid, myeloid, and megakaryocytic precursor populations present. In six CKO and two HET long bones examined, multiple foci of bony or cartilaginous growth extended into the medullary cavity or protruded beneath the periosteum (Figure 4C,D).

### 3.3. Osteochondral Dysplasia in Axial Skeletons of CKO Mice

In eight CKO and two HET mice, the cartilage at the ends of the sternebrae was minimally to mildly increased and the bones were subtly atypical, with mildly thinned cortices with irregular margins suggestive of delayed endochondral ossification. One HET mouse had significant and grossly visible dysplasia of multiple ribs, resulting in a beaded appearance due to the expansion of cartilage with minimal transition to bone (Figure 4E,F). In contrast to the lesions observed in long bones, the vertebrae in CKO mice were smaller than those of Cre- mice but lacked changes consistent with osteochondral dysplasia. The cortices were thin with less bone content, but we did not observe cartilaginous or fibrous connective tissue segments. Some intervertebral discs were small in CKO mice (Figure 5), while others were closer in proportion to those in Cre- mice. All CKO and HET mice had these vertebral abnormalities.

Osteochondral lesions and defects in development were also found in the calvaria of all nine CKO mice, with multifocal presentation in the frontal, parietal, interparietal, and occipital bones. These changes included reductions in bone matrix and changes similar to those identified in long bones. Particularly in the interparietal and occipital bones, affected calvarial regions contained large irregular stretches of cartilage. When bones or cartilage were lacking, the meninges were overlain by variable amounts of fibrous connective tissue and large numbers of round to oval proliferative cells suspected to be a proliferative precursor population. Skull sutures were larger than expected in eight CKO and three HET mice and contained large numbers of similar precursor cells (Figure 6).

### 3.4. Facial Abnormalities in CKO Mice

The teeth in all examined sections, especially the maxilla and mandible molars, demonstrated a wide variety of phenotypes (Figure 7). The majority of incisors remained within normal limits, while molars were significantly altered in shape and/or unerupted and covered by oral mucosa. Maxillary and mandibular molars demonstrated short tooth roots and varied from elongate and irregularly pointed to scalloped, smaller, and more compressed than comparable Cre- mice. Tooth roots were frequently poorly seated in the jawbones, which had changes similar to those present in other osteochondrodysplastic bones. In eight CKO mice and one HET mouse, both maxillary and mandibular apposed teeth were abnormal, and in one CKO mouse and three HET mice, teeth were asymmetrically developed, with relatively normal teeth opposite underdeveloped teeth. In three CKO mice, 1–2 incisor teeth lacked mineralized tissue, normal cellular differentiation, and structure. These teeth were instead composed of numerous concentric layers of large proliferative precursor cells that vaguely formed the shape of a tooth. A single tooth in one CKO mouse exhibited inappropriate anatomic localization, occurring as a unilateral primitive tooth-like structure in the lateral nasal gland lumen surrounded by dysplastic bone. There were inconsistencies in the number and appearance of odontoblasts in all examined postnatal ages, with corresponding irregularities in the dentin. In six examined molars from five CKO mice, odontoblasts were segmentally replaced by necrotic cellular debris, and odontoblasts were absent from the roots of some teeth. In seven CKO and three HET mice, a large pocket was present beneath the oral mucosa and contained a small bud-like structure resembling a small tooth, but this lacked typical lamellar organization. These pockets were frequently filled with eosinophilic material. Soft tissue injuries, including lacerations, erosion, hemorrhage, inflammation, and compensatory epithelial hyperplasia were common around unerupted molars, occurring in eight CKO and three HET mice. Unsurprisingly, bony lesions consistent with osteochondral dysplasia were present in the jaw.

In addition to a reduced head size, the CKO mice had markedly shortened noses, with occasional lateral deviation and misalignment. The nasal turbinates and cribriform were irregular and exhibited similar developmental defects as other bones in the CKO mice (Figure 7). The bones were irregularly thin with uneven cortices, irregular multifocal pockets of immature, deeply basophilic cartilage, and variable marbling in the bone. The ethmoid turbinates of seven CKO mice and one HET mouse were surrounded by neutrophils that infiltrated through the lamina propria and entered the olfactory epithelial layer. The cribriform was irregular and affected by similar bony changes in seven CKO and two HET mice and was also lined by neutrophils in eight CKO and two HET mice. Some nasal turbinates were increased in size and irregular due to the proliferation of fibrous connective tissue, resulting in a larger profile than was expected. Lesions were observed in the turbinates of nine CKO and three HET mice. Similar lesions were noted in the nasal septum of seven CKO and two HET mice, with the junction of the distal septum and palate frequently consisting of fibrous connective tissue rather than bone. If bone was present, it was often reduced and irregular, resulting in deviation of the septum. In four CKO mice and one HET mouse, the hard palate was variably composed of fibrous connective tissue, irregular dysplastic bone, irregular reduced cartilage, and oval precursor cell population similar to that seen in the calvaria. The mucosal epithelium and fibrous connective tissue of the palate were intact in all examined mice. The size and shape of the lateral nasal glands and sinuses were altered in nine CKO and three HET mice because of bony abnormalities and replacement by fibrous connective tissue. In five CKO mice and one HET mouse, the lumens of these structures contained flocculent eosinophilic material and inflammation varying from macrophages to admixed macrophages and neutrophils. One CKO mouse and one HET mouse also had alterations of the epithelium of the nasopharyngeal meatus, with apical blebbing and luminal neutrophilic inflammation admixed with mucus. The olfactory sensory epithelium found in the caudal nasal passages at the level of the ethmoid turbinates was attenuated for six CKO mice and one HET mouse. This change corresponded to disorganization and disruption of cells in the olfactory bulb glomerular layer in the same six CKO mice and one HET mouse, recapitulating the findings of a previous study indicating off-target Cre expression at this location [2] (Figure 2G). The tympanic bullae and cochleas of many CKO mice also had lesions consistent with osteochondral dysplasia, although the overall shape and size of these structures was not altered compared to Cre- control mice. CKO mice with neutrophilic inflammation along the nasal turbinates and cribriform had similar inflammation in the external and middle ears. Two CKO mice with neutrophilic inflammation in the nose had similar inflammation in the tunica muscularis of the stomach, and in one of these mice a small number of neutrophils was also present in the muscularis mucosa and between Paneth cells in the mucosa.

## 4. Discussion

TGFβ signaling has been demonstrated to be crucial to skeletal development in multiple models. The Cre/loxP system has been used extensively to generate conditional gene deletion mice, including those with deletion of Tgfbr2 [6,7,24,25,26,27,29,33,34,35]. Several of these mouse models are embryonic or perinatal lethal [24,25,26,27,29,33,34]. Osterix-Cre mice have been crossed with mice containing floxed alleles of Tgfbr2 to delete Tgfbr2 specifically in immature osteoprogenitors. While the embryonic lethality with Tgfbr2 deletion earlier in development could be linked to severe skeletal defects, such as failure to form a skull [24,25], there was no clear cause of death in mice with the Osterix-Cre-driven Tgfbr2 deletion. We attempted to prolong the life of CKO mice with mash and nutritional supplements, but these mice never reached full adulthood, and only occasionally survived past 60 days before demonstrating progressive signs of weakness followed by death. Additional tests would be required to confirm any differences in strength and to rule out whether pain related to bone and joint abnormalities contributed to weakness of movement. Because no clear and direct cause of death was evident during our investigations, we present a data trend from the pathological analyses suggesting gross skeletal deformities likely contributed to postnatal deaths. Osterix-Cre recombination is also available in an estrogen receptor ligand-binding domain (Osterix-Cre-ERT) that allows temporal induction with administration of tamoxifen or 4-hydroxytamoxifen [36]. A parallel evaluation using the Osterix-Cre-ERT has not yet been performed and would provide useful information about the role of Tgfbr2 in supporting bone development into adulthood.

Interestingly, we found demonstrations of osteochondral dysplasia throughout the skeleton, including the long bones, ribs, and skull. This abnormal phenotype bears resemblance to the skeletal phenotype demonstrated when Tgfbr2 was deleted earlier in development using Prx1-Cre;Tgfbr2^f/f^. The Prx-1 knockout animals demonstrated a delayed/halted hypertrophic differentiation in the growth plates, leading to reduced bone lengths similar to our mice. None of our mice exhibited the joint fusion demonstrated in the Prx1-Cre;Tgfbr2^f/f^ mice [24,25], but we did see disrupted rib development, as well as thick stretches of cartilage and fibrous connective tissue in place of bone throughout the skeleton. Mice with deletion of Tgfbr2 in Col2A-Cre-expressing cells demonstrated malformations in the base of the skull, which normally undergoes endochondral ossification [36]. These findings were also similar to our CKO mice, but the Col2A-driven Tgfbr2 deletion did not affect chondrocyte differentiation or intramembranous differentiation [34], unlike our CKO mice. However, deletion of Osterix in Col2A-Cre:Osterix^f/f^ mice severely impaired chondrocyte differentiation and endochondral ossification, also resulting in perinatal lethality [37]. Interestingly, inducible deletion of Osterix using the Col2-Cre-ERT model impaired secondary ossification in the epiphyses in a manner remarkably similar to our CKO mice. These findings suggest that the chondrocyte hypertrophy and osteoblast differentiation during secondary ossification in postnatal bone development are driven by Osterix [2]. Our conditional deletion of Tgfbr2 was driven by the Osterix promoter, suggesting that Osterix regulates secondary ossification via TGFβ signaling. Most strikingly, our CKO mice are, to the best of our knowledge, the only example in which intramembranous ossification of the calvaria was replaced with large patches of fibro-cartilaginous tissue. We propose that Osterix regulates an endochondral ossification switch through TGFβ signaling that can be highly manipulated during late embryogenesis and postnatal stages to produce skeletons with less mineralized tissue and a more chondrogenic skeleton, similar to those reported in previous studies [38,39].

In addition to the striking chondrogenic calvaria, we found fibrous connective tissue and proliferative precursor cells in place of the bony hard palate (Figure 7 and Appendix A) in some CKO mice. This likely contributed to the altered nose and facial structures and correlates with previous studies of the role of TGFβ signaling in clefting [26,27,40,41,42,43,44,45] despite the absence of a cleft demonstrated in this model (Appendix A). Cleft palate disorders can be caused by a variety of signaling disruptions in development, which have been outlined in reviews [43,46]. To date, these models have focused on TGFβ deletion earlier in development [47] and have not investigated palatal disruptions when TGFβ signaling is abrogated later, as is the case with the Osterix-Cre mouse model. Further investigations into TGFβ abrogation later in skeletal development could provide information relevant to skeletal diseases initiated later in life. Based on our mouse skeletal phenotype, we suggest that our CKO mice may provide a good model to investigate how TGFβ signaling contributes to the craniosynostosis and oro-dental anomalies demonstrated in Loeys-Dietz syndrome [44,48,49,50,51,52]. Future investigations should include in-depth static and kinetic morphometric analysis building upon our previous publication demonstrating rigorous quantification of histomorphometric changes in the CKO long bones, growth plates, and calvaria [6].

During our investigations aimed to ascertain the cause of death in the CKO mice, we performed extensive pathological analyses of the entire bodies of the mice and used ROSA26-mTmG reporter mice to confirm the possible presence of off-target Cre recombinase activity. Histopathological analyses revealed that there were no significant defects in non-skeletal organs outside of the olfactory bulb, which was previously reported as having off-target Cre expression [23]. However, there was minor Cre expression detected in the stomach, intestines, liver, kidneys, and lungs. The differences in these tissues, combined with the observed skeletal abnormalities, might have led to changes in breathing, feeding, digestion, and/or metabolism. This led us to conclude that the most likely cause of death was malnutrition and starvation. The abnormal tooth morphologies most likely affected feeding behavior, and the timing of the deaths correlated with the failed tooth eruption that could not facilitate the transition from milk to solid food consumption. Although we provided food softened into mash, it was likely still difficult for the mice to chew with their misaligned teeth, which might have also been hypermobile due to the short roots that failed to properly anchor the molars into the alveolar bone. The presence of necrosis, inflammation, and compensatory hyperplasia in the oral mucosa surrounding the abnormal teeth suggests repeated trauma to these areas, likely resulting in pain with chewing and a reluctance to continue masticating food pellets. In addition, there were defects in the nasal turbinates, including variable bony distortion at all examined levels of the nose. These resulted in shortening and compression of the nasal passages, as well as multifocal attenuation of the olfactory sensory epithelium. Combined with the disorganization of the olfactory bulb glomerular layer, this may have disrupted the ability of the mice to smell and taste, further affecting feeding behaviors. Interestingly, the ability to smell can also change metabolism and autonomic innervation, leading to decreased weight without changes in food consumption [53]. Further studies would be required to assess the impact on smelling/tasting, as well as the specific impact on feeding behaviors and metabolism.

As noted above, we also observed off-target expression of Cre recombinase in the stomach and intestine [23], as well as previously unpublished diffuse expression in the liver, lungs (Figure 2H,K), and kidneys. This off-target Tgfbr2 deletion throughout the gastrointestinal tract could have affected intestinal stem cell differentiation [54] and, consequently, the ability of the mice to absorb nutrition and obtain calories. However, we did not find any histopathological abnormalities in these organs despite detection of off-target Cre recombinase expression in the stomach and intestine. Additional studies would be required to determine if metabolic and/or absorption abnormalities contributed further to weight loss.

There was striking neutrophilic inflammation along the nasal turbinates, the cribriform, and in the ears in a subset of HET and CKO mice. There was also significant necrosis in the teeth and oral cavity, which could have contributed to a localized neutrophilic inflammatory response. The reasons for inflammation immediately along the nasal turbinates and cribriform are less clear. In some areas of the nose, there were small foci of necrosis in the lamina propria, which may account for some of these infiltrates. Similarly, in five CKO mice, the lateral nasal glands were multifocally necrotic, which may have also contributed to the orientation of many neutrophils along bony structures. The turbinate- and cribriform-oriented nature of much of the inflammation could be explained by disruptions in TGFβ-regulated chemokine signaling, which regulates neutrophil chemotaxis and behavior, as has been published in other unrelated models [55,56,57,58,59,60]. Similar inflammation was present in the stomachs of two CKO mice without obvious causative lesions and may be due to similar chemotaxis pathways. While the neutrophilic inflammation was not severe at any one location, this also may have contributed to weight loss via a mechanism such as cachexia.

Since there was no off-target Cre recombinase activity in the heart, the lesions in the heart must be due to indirect causes or due to a systemic effect of the off-target Cre expression we found in the other organs. It is well known that starvation can lead to heart damage [61,62,63,64], including myocardial necrosis [65]. It is possible that the hearts in the CKO and HET mice demonstrated necrosis due to malnutrition/starvation. The cause of the multinucleated myofibers in the diaphragms and limb-associated muscles of CKO mice is unclear. This morphology is most commonly associated with a regenerative response following injury, but we did not observe signs of skeletal myonecrosis, nor were there signs of inflammation to indicate previous injury. However, the enlarged ribs and deformed nasal passages suggest these mice may have experienced compromised breathing mechanics, including constricted air movement, which could have contributed to the abnormal phenotype we found in the diaphragm and caused the demonstrated heart injury [66,67,68]. If the constricted airflow caused hypoxia, this could have additionally led to the myocardial lesions similar to those demonstrated in a more severe model of induced myocardial damage using normocapnic hypoxia in piglets [69]. While we did find off-target Cre expression in rare, isolated cells of the lungs, we did not find any phenotypic alterations in the lungs that suggest a defect. Quantitative and functional investigations would be necessary to rule out the hypothesis that lung defects contributed to respiration difficulties in CKO mice. Further testing is recommended to ascertain whether these mice were hypoxic and if that contributed to their weakened states and early deaths.

## 5. Conclusions

In conclusion, we highlighted several skeletal elements where bone development was dysregulated by the deletion of Tgfbr2 in osteoprogenitor cells using the Osterix-Cre;Tgfbr2^f/f^ mouse model. We did not find any distinct organ failures that correlated with off-target Cre expression that were responsible for the postnatal morbidity. Instead, we herein provide evidence that there were several concurrent organ abnormalities, primarily in the skeletal tissues, that we believe led to weakened states and fatal malnutrition and starvation. Further investigations into craniofacial structure development, as well as the osteogenic-to-chondrogenic fate change throughout the CKO skeletons, will include more in-depth quantifications to best evaluate the significance of TGFβ signaling in bone development. Such studies stand to provide therapeutic insight into skeletal diseases associated with similar phenotypes.

## Figures and Tables

**Figure 1 genes-12-00975-f001:**
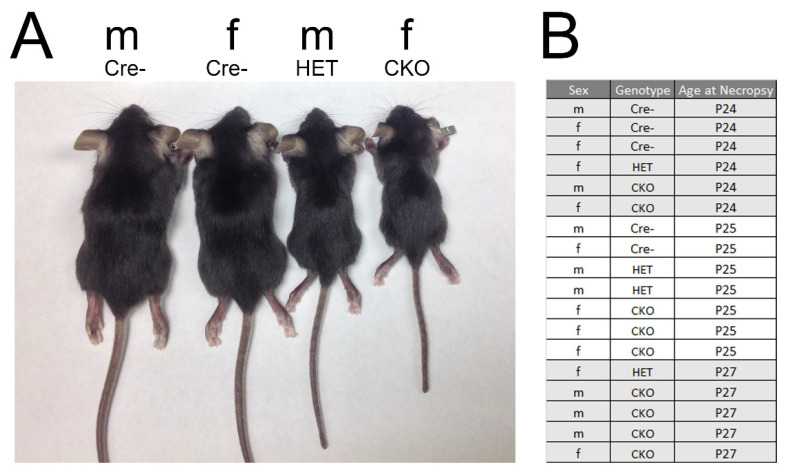
Representative mice used in the pathology analyses. (**A**) Images of male (m) and female (f) Cre-, HET, and CKO mice. (**B**) The sex, genotype, and age of mice used in the evaluation. Five Cre-, four HET, and nine CKO mice were analyzed. M: male; F: female.

**Figure 2 genes-12-00975-f002:**
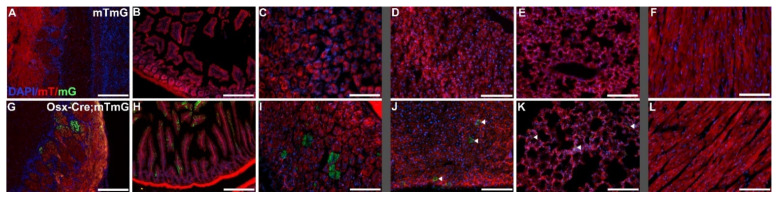
Demonstration of Cre recombinase activity driven by the Osterix-GFP-Cre promoter. Select organ sections of Osterix-Cre-;ROSA26mTmG mice (mTmG; (**A**–**F**)) and Osterix-Cre+;ROSA26mTmG mice (Osx-Cre;mTmG; (**G**–**L**)) showing fluorescence of membranous tdTomato (mT) in the absence of Cre recombinase activity and membranous EGFP (mG) in the presence of Cre recombinase activity. Widely expressed off-target Cre recombinase was confirmed in the olfactory bulb glomerular layer (**G**), the intestine (**H**), and the stomach (**I**). Sparse off-target Cre recombinase expression was seen in the liver (**J**) and the lungs (**K**), designated by arrows. No Cre recombinase activity was present in the heart (**L**). Scale bars (**A**,**B**,**G**,**H**) = 200 µm. Scale bars (**C**,**F**,**I**,**L**) = 100 µm.

**Figure 3 genes-12-00975-f003:**
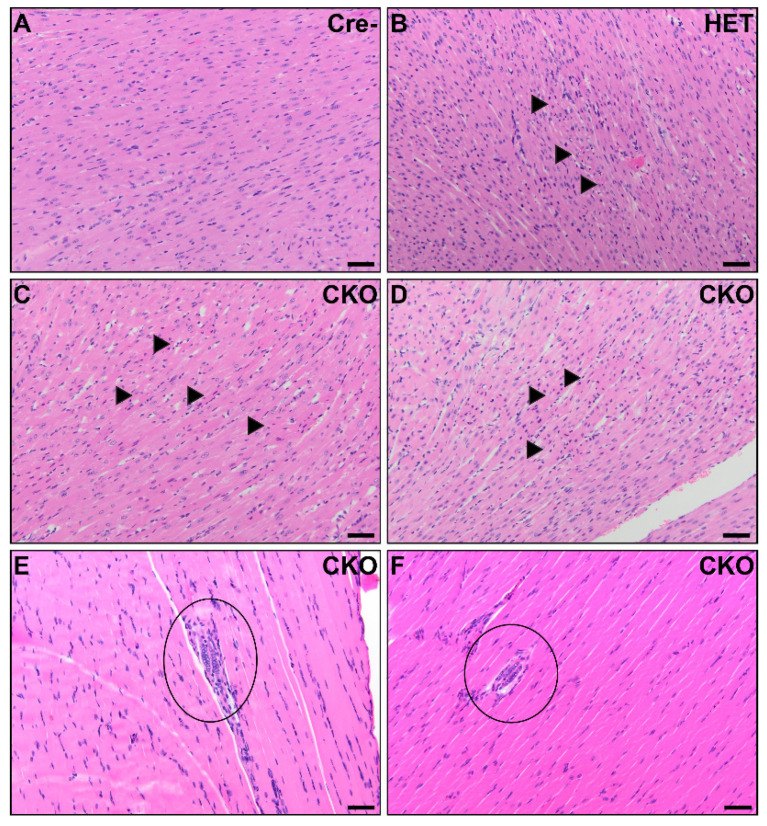
Cardiac and skeletal muscle pathology in CKO mice. In the myocardium, small foci consistent with acute myocardial necrosis were present in CKO mice (**C**,**D**) and one HET mouse (**B**). Minimal inflammation was present around the hypereosinophilic, disrupted, and disorganized cardiomyocytes with hyperbasophilic shrunken nuclei (arrows). No significant microscopic lesions were detected in the myocardium from Cre- age-matched control mice (**A**). There were multinucleated skeletal myofibers present in the diaphragm of two CKO mice (circles). Four CKO mice had similar multinucleated myofibers in perivertebral skeletal muscles. The skeletal muscle in the diaphragm was unremarkable in Cre- control mice. Scale bars (**A**–**F**) = 50 μm.

**Figure 4 genes-12-00975-f004:**
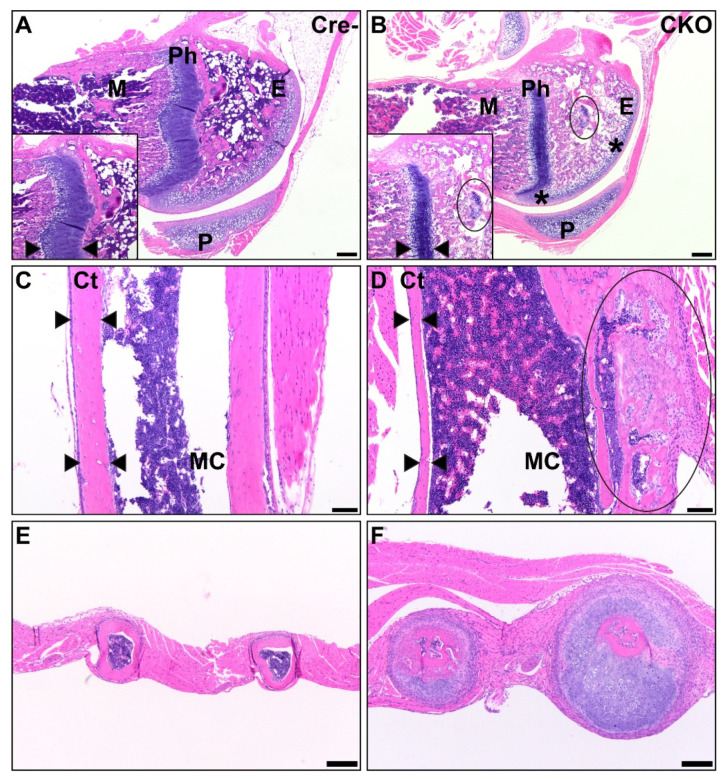
CKO mice exhibit osteochondral dysplasia in long bones (**B**,**D**,**F**) compared to Cre- age-matched controls (**A**,**C**,**E**). A/B: In the femur, there was thin and sclerotic articular cartilage (asterisk), sparse epiphyseal trabeculae with retained cartilage cores (circle), thin and disorganized physes, and decreased metaphyseal primary spongiosa present in CKO mice (**B**) compared to control mice (**A**). Insets show a comparison of the physis thickness, with margins demarcated by arrows, and the retained cartilage core. Scale bars = 200 µm. E: epiphysis, Ph: physis, M: metaphyseal primary spongiosa, P: patella. C/D: In the humeral diaphysis, CKO mice (**D**) had markedly thinner cortices compared to age-matched Cre- control mice (**C**). CKO mice had occasional subperiosteal growths (circle) of cartilage, woven bone, remodeling cortical bone, and precursor-like cells. The cortex thickness is delineated by arrows. Scale bars = 100 μm. Ct: cortex, MC: medullary cavity. E/F: The ribs of a CKO mouse (**F**) were many times larger than those of a comparable Cre- control mouse (**E**) due to excessive cartilage surrounding a central core of irregular bone. Scale bars = 200 μm.

**Figure 5 genes-12-00975-f005:**
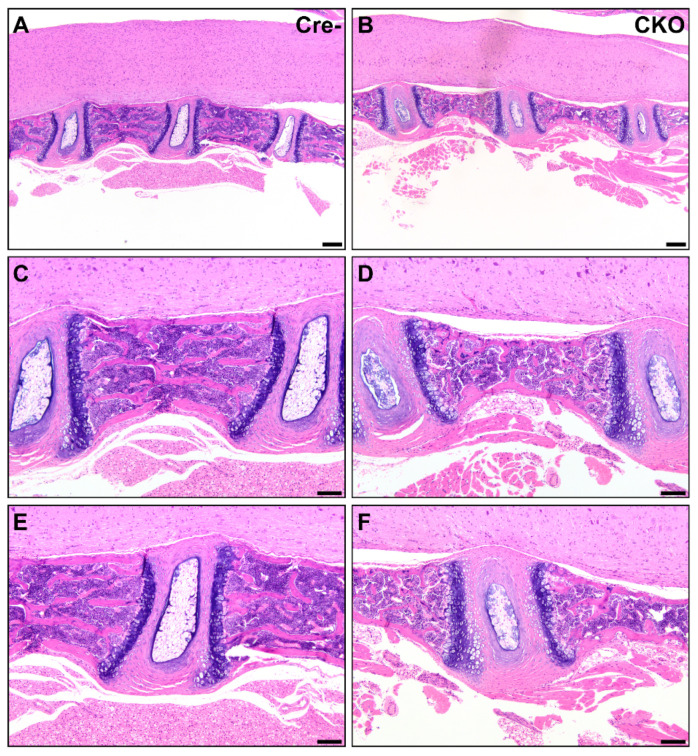
Reduced soft and mineralized tissue in CKO vertebrae. CKO mice (**B**,**D**,**F**) had smaller vertebrae and intervertebral disks than Cre- age-matched control mice (**A**,**C**,**E**). Changes similar to those found in other bones were present in the vertebrae from CKO mice (**D**), with thinned cortices and thin trabeculae compared to comparable Cre- control mice (**C**). The intervertebral disks were also smaller in CKO mice (**F**) but had similar differentiation of the nucleus pulposus and annulus fibrosus. The cartilage end plates were comparable. Scale bars (**A**,**B**) = 200 μm. Scale bars (**C**–**F**) = 100 μm.

**Figure 6 genes-12-00975-f006:**
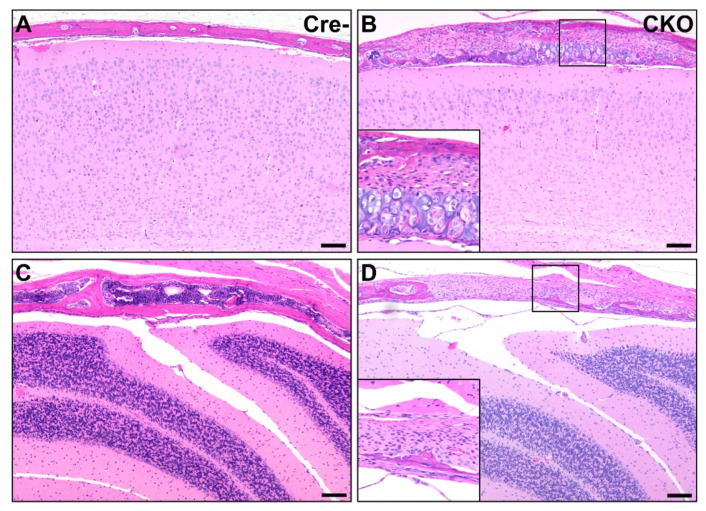
Osteochondral dysplasia in multiple bones of the calvaria in CKO mice. Compared to Cre- age-matched control mice (**A**,**C**), multiple bones were affected in CKO mice, including the parietal bone (**B**) and interparietal bone (**D**). Lesions similar to the osteochondral dysplasia observed elsewhere and consisting of variable amounts of bone, cartilage, fibrous connective tissue, and precursor-like cells were present in some areas (**B**), while in others, the bone was segmentally absent, with the brain encased only by fibrous connective tissue and precursor-like cells (**D**). Insets in (**B**,**D**) provide higher magnification views. The photos shown in (**A**,**B**) were taken mid-parietal at the level of the middle hippocampus, while (**C**,**D**) were taken mid-interparietal over the cerebellum. Scale bars = 100 μm.

**Figure 7 genes-12-00975-f007:**
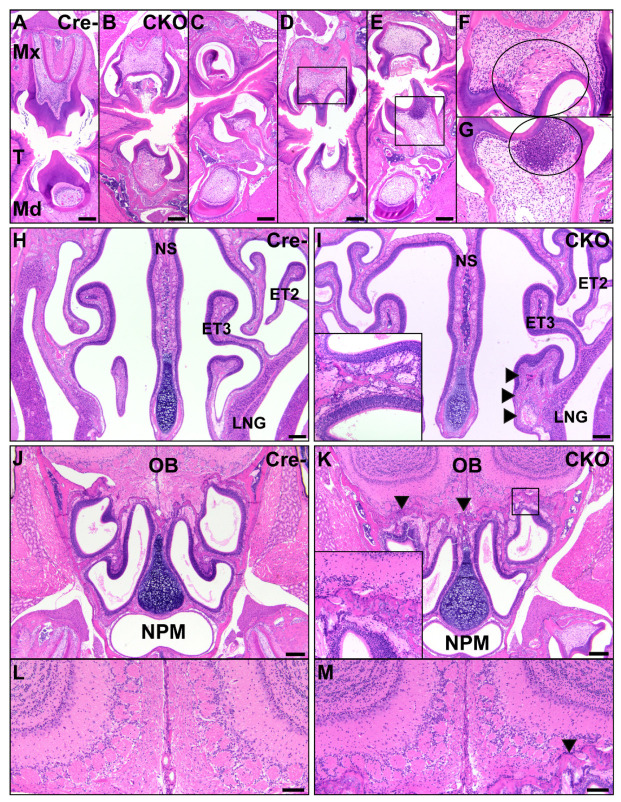
Abnormal facial structures in CKO mice. The CKO mice (**B**–**G**) had numerous abnormalities of tooth development in the maxillary and mandibular teeth, particularly in molars, compared to Cre- control mice (**A**). These included delayed eruption (**B**,**C**), short tooth roots (**B**–**E**), underdevelopment of the tooth (**C**), and necrosis of odontoblasts and the pulp cavity with infiltration of neutrophils (**D**,**E**). A higher magnification of the pulp cavities shows necrosis with clefting ((**F**), circle) and a marked neutrophilic response to necrosis ((**G**), circle). Scale bars (**A**–**E**) = 200 μm. Scale bars (**F**–**G**) = 50 μm. T: tongue; Mx: maxillary; Md: mandibular. In the nasal cavity, the CKO mice (**I**) had consistent osteochondral dysplasia of the nasal turbinates, with variations in the affected bones and involvement of the nasal septum, compared to Cre- control mice (**H**). The inset shows the presence of neutrophilic inflammation lining a nasal turbinate from a different CKO mouse. Neutrophils extended to the cribriform in CKO mice but were not present in Cre- controls. The arrows delineate a dysmorphic, enlarged ethmoid turbinate. Scale bars = 200 μm. NS: nasal septum; ET2: ethmoid turbinate 2; ET3: ethmoid turbinate 3; LNG: lateral nasal glands; OB: olfactory bulb; NPM: nasopharyngeal meatus. Compared to Cre- controls (**J**,**L**), CKO mice (**K**,**M**) had changes consistent with osteochondral dysplasia of the cribriform with neutrophilic infiltrates along the bone. The arrows indicate the margins of the osteochondral tissue. The inset is a higher magnification image of the osteochondral tissue. CKO mice (**M**) had disorganization of the olfactory bulb glomerular layer compared to Cre- controls (**L**). The arrow indicates a small region of neutrophilic infiltration along the cribriform. Scale bars = 100 μm.

## Data Availability

All data we have is shown in this paper unless otherwise noted. Additional images are available upon request.

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
