# Peer review of "Skeletal Deformities in Osterix-Cre;Tgfbr2f/f Mice May Cause Postnatal Death"

_genes, 2021, doi:10.3390/genes12070975_

Round 1

Reviewer 1 Report

In this study, Corps et al. describe a number of skeletal deformities in a mice model of Tgfbr2 loss in Osterix-expressing cells and speculate on the cause of death of these animals. The study is well structured and executed, but mostly focuses on the skeletal abnormalities of the animals. However, as the title suggests, the study would be related to the cause of the death which, so far, seems still speculative Thus, I fail to understand the overall significance and novelty of the study. In my view, either:

  1. the study would be focused on the description of the phenotype, the title would be changed and the manuscript would be restructured accordingly.
  2. the study would maintain its focus on the cause of death, but the relevance should be clearly stated and an effort into having more conclusive, rather than speculative, findings should be made.

Technically, the study seems to be mostly well-executed and scientifically correct.

However, I highlight some points:

  1. Although Cre recombinase was not observed in the heart, the authors did not consider the potential systemic effects of off-target Cre expression on myocardial necrosis. I feel this to be especially relevant, considering the leaky nature of this genetic reagent in many different organs
  2. It is unclear how many animals were used in each analysis
  3. Many scientifically inaccurate terms/statements are found throughout the text. Examples: “In some cases…”, “…the CKO mice frequently had…”, “were frequently surrounded by…”, “…it was often reduced and irregular”. It would help to have numbers to accompany these phenotypic descriptions
  4. Also, when/if possible, quantifications of the skeletal defects would help to evaluate the significance of these findings. As an example, the authors mention that “…the vertebrae in CKO mice were smaller than those of Cre- mice…”, but vertebral size does not seem to be measured

Reviewer 2 Report

The manuscript titled “Postnatal death in Osterix-Cre;Tgfbr2f/f mice may be related to skeletal deformities” represents an extensive pathological analysis and comparison between Osterix-Cre-, Het and CKO mice.  The potential off target Cre expression is also examined by crossing the Osterix-Cre mice with the ROSA26mTmG reporter line.  The paper is well-written and it confirms and expands upon previous findings of skeletal and tooth abnormalities by demonstrating osteochondral dysplasia throughout the appendicular and axial skeleton of CKO mice including the craniofacial skeleton.  This is an important and interesting study, and will hopefully lead to a more comprehensive histomorphometric analysis of the cartilage and bone abnormalities.  Specific comments to be addressed by the authors include;

  • Figure 3 figure legend, second sentence. In the myocardium, small foci …   
  • Throughout the results, the authors refer to the mineralization status of the bone (e.g. hypomineralization). Since the samples are demineralized prior to sectioning and staining, what they are actually referring to is the amount of bone (organic matrix) that is present in the sections, not the level of mineralization of the bone.  To draw any conclusions about the mineralization would require sectioning and staining of plastic-embedded, undemineralized samples.  The authors should change the terminology used whenever referring to the amount of bone in the CKO compared to Cre-
  • The authors provide a semi-quantitative assessment of the amount of bone, the thickness of cortical bone or the growth plate, or the number of osteoblasts. There is no quantitative assessment of any of these parameters provided.  Although many of the changes described are clearly visible, this model would benefit from a more comprehensive histomorphometric assessment of the defects in cartilage, bone and the craniofacial skeleton.  Perhaps future studies could provide static and kinetic morphometric analysis in collaboration with an experienced skeletal histomorphometrist.  It is difficult to support the result “lined by fewer osteoblasts than expected” without providing direct (quantitative or visual) evidence in support of this statement.
  • In figure 7, there is no image showing the palate or the junction of the inferior part of the nasal septum with the palate. 7H and I are cut off at the inferior end of the nasal septum and do not include the palate. It does not look like the nasal septum connects with the palate though it is difficult to know for sure since it is cut off.  In the results it states “…with the junction of the distal nasal septum and the palate frequently consisting of fibrous connective tissue rather than bone”.  The figure does not allow one to visualize this finding.  Furthermore, the results state that “Occasionally, the hard palate was almost entirely composed of fibrous tissue in CKO mice.”  Once again, these important data are not shown in figure 7 as alluded to in the text. It was surprising that there was no mention of cleft palate in these CKO mice given the important role of TGF-b signaling during palate development.  At the very least, a section to demonstrate what the palate looks like in the CKO mice as well as a discussion regarding the presence or absence of cleft palate in these mice would be helpful.
  • Given the grossly enlarged ribs and the more subtle defects in the skeletal muscle of the diaphragm, it is surprising that the possibility of a respiratory defect involving the mechanics of breathing (inspiration and expiration) was not even mentioned as a possible cause of death in these mice. The normal expansion of the pleural cavities upon inspiration would appear to be compromised given the grossly enlarged ribs.  Perhaps this possibility should also be further explored and considered as a contributing cause of premature death in the CKO mice.   

Round 2

Reviewer 1 Report

The authors have addressed major concerns.

Author Response

Thank you for your comments.